# A Global Gridded Dataset for Cloud Vertical Structure from Combined CloudSat and CALIPSO Observations

William Bertrand[1,2], Jennifer E. Kay[1,2], John Haynes[3], and Gijs de Boer[2,4]

[1]Department of Atmospheric and Oceanic Sciences (ATOC), University of Colorado Boulder, Boulder, CO, USA
[2]Cooperative Institute for Research in the Environmental Sciences (CIRES), University of Colorado Boulder, Boulder, CO, USA
[3]Cooperative Institute for Research in the Atmosphere (CIRA), Colorado State University, Fort Collins, CO, USA
[4]National Oceanic and Atmospheric Administration, Physical Sciences Laboratory, Boulder, CO, USA

**Correspondence:** William Bertrand (william.bertrand@colorado.edu)

**Abstract.** The vertical structure of clouds has a profound effect on the global energy budget, the global circulation, and the atmospheric hydrological cycle. The CloudSat and Cloud-Aerosol Lidar and Infrared Pathfinder Satellite Observations (CALIPSO) missions have taken complementary, colocated observations of cloud vertical structure for over a decade. However, no globally-gridded dataset is available to the public for the full length of this unique combined data record. Here we present

the 3S-GEOPROF-COMB product, a globally-gridded (level 3S) community data product summarizing geometrical profiles (-GEOPROF) of hydrometeor occurrence from combined (-COMB) CloudSat and CALIPSO data. Our product is calculated from the latest release (R05) of per-orbit (level 2) combined cloud mask profiles. We process a set of cloud cover, vertical cloud fraction, and sampling variables at 2.5, 5, and 10 degree spatial resolution and monthly and seasonal temporal resolution. We address the 2011 reduction in CloudSat data collection with Daylight-Only Operations (DO-Op) mode by subsampling pre-

2011 data to mimic DO-Op collection patterns, thereby allowing users to evaluate the impact of the reduced sampling on their analyses. We evaluate our data product against CloudSat-only and CALIPSO-only global-gridded data products as well as four comparable surface-based sites, underscoring the added value of the combined product. Interest in the product is anticipated for the study of cloud processes, cloud-climate interactions, and as a candidate baseline climate data record for comparison to follow-up satellite missions, among other uses.

## 1 Introduction

The vertical structure of clouds fundamentally impacts and expresses the global circulation (Mace et al., 2007; Stephens et al., 2002), the atmospheric hydrological cycle (Stephens et al., 2002), and the global energy budget (Henderson et al., 2013; Oreopoulos et al., 2017). The cloud response to climate change is a major driver of uncertainty in climate predictions (Sherwood et al., 2020), and global measurements of the vertical structure of clouds can improve understanding of cloud-

climate feedbacks. While numerous passive satellites detect clouds, cloud vertical structure is most directly retrieved with active remote sensing. CloudSat and CALIPSO, space-borne radar and lidar (Marchand et al., 2008; Winker et al., 2010), have taken collocated active remote sensing observations of cloud vertical structure. Their complementary measurements

provide the first decade-plus climatology of cloud vertical structure. Many combined data products exist at the individual orbit level (level 2), and CloudSat and CALIPSO both have single-instrument global gridded (level 3) products (Haynes, 2020; NASA/LARC/SD/ASDC, 2018, 2019), but a combined level 3 data product has not been produced, peer-reviewed, and distributed to the public. Here, we present the level 3S hydrometeor GEOmetrical PROFile COMBined (3S-GEOPROF-COMB) CloudSat+CALIPSO product, a comprehensive globally gridded combined product for 2006-2020. This new product is needed because a temporally-aggregated, globally-gridded, combined level 3 product provides tremendous value for global change researchers.

## 1.1 Comparison of complementary instrument capabilities

While CloudSat and CALIPSO both actively measure hydrometeors through the atmospheric column, CloudSat's milimeter-wavelength radar (94 GHz, 3.2 mm) and CALIPSO's nanometer-wavelength lidar (532/1024 nm) have uniquely different and complementary atmospheric profiling capabilities. When taken together, these two instruments provide a more comprehensive measurement of cloud vertical structure than either would on its own. First, both radar and lidar measure returned backscatter from the atmospheric column, but the two instruments attenuate differently. Due to the lidar's shorter wavelength, scattering layers with small particle size and/or low optical thickness (e.g. aerosol or cirrus cloud layers) will have a stronger return for the lidar than the radar. While this increased sensitivity allows the lidar to detect thin cloud and aerosol layers, it also means that optically thick layers attenuate the lidar and prevent measurement below the altitude of attenuation Liu et al. (2022). In contrast, while the radar does not detect optically thin layers or small droplet sizes, it only attenuates in the most extreme of precipitation events (∼0.3% of profiles (Mace et al., 2007)). Second, the CloudSat radar has 'surface clutter' preventing measurement in the lowest 500 m of the atmosphere (Marchand et al., 2008), whereas CALIPSO's lidar allows measurement of clouds near the surface (Winker et al., 2009). Through the combination of both instruments, the only situation when a full vertical column of cloud observation is not obtained is in the lowest 500-1000 m of the atmosphere when the lidar is attenuated and the radar is obstructed by surface clutter.

An example of CloudSat+CALIPSO synergy in detecting clouds can be seen in a combined radar-lidar cloud mask for a segment of a single orbit in Figure 1. In the optically thick precipitating systems, both the radar and the lidar detect the top of the cloud, but after a few kilometers the lidar attenuates and stops detecting cloud. In this case, the radar fills in the lidar's data gap. On the other hand, scattered low clouds <1 km are lidar-only, since they lie in the radar surface clutter region. In this case, the lidar fills in the radar's data gap. For a range of scenarios, the two instruments fill in each other's data gaps for a more comprehensive measurement combined than separately.

While these differences in measurement capability between CloudSat and CALIPSO affect cloud detection in individual cloud scenes, they also impact the globally aggregated map of hydrometeors. Figure 2 shows 2006-2011 zonal-mean vertical cloud fraction, comparing CloudSat's level 3 cloud product (Haynes, 2020) to CALIPSO's level 3 cloud product (NASA/LARC/SD/ASDC, 2018). While both datasets capture roughly the same pattern, the shape and magnitude of the global distribution of clouds differs due to the aforementioned instrument capabilities. For example, CloudSat (Fig. 2a) has 10-15% more equatorial mid-level (3-6 km) cloud fraction than CALIPSO (Fig. 2b) due to thick, deep convective clouds attenuating the lidar.

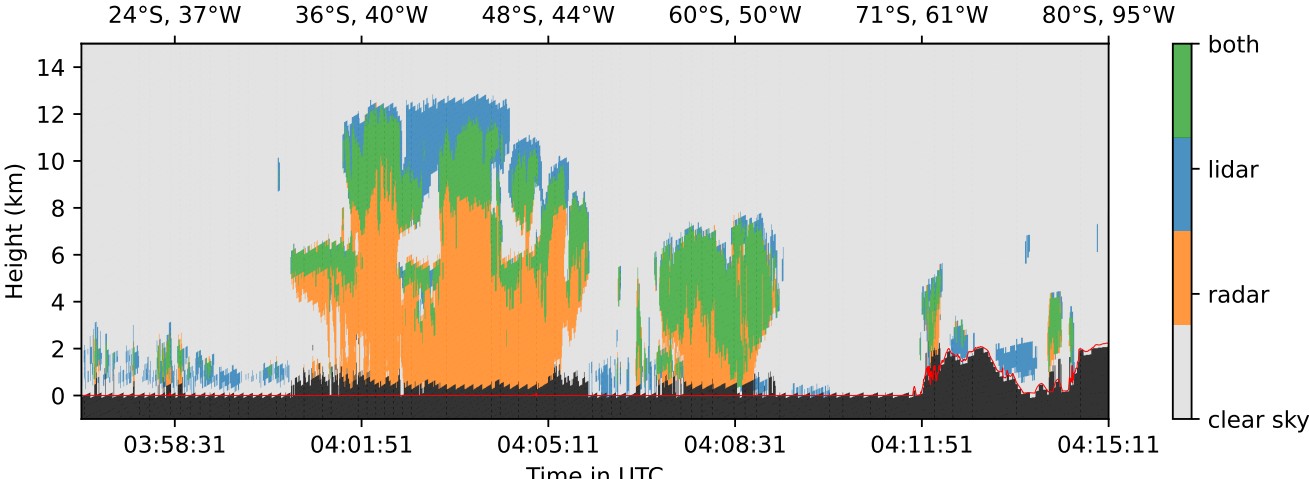

**Figure 1.** Sample quicklook of a snippet of granule 23108, collected on 1 September 2010. The figure shows collocated radar and lidar cloud masks, indicating regions detected by lidar only, radar only, or lidar and radar. Black pixels indicate no measurement (subsurface data or surface clutter), while the red trace indicates the height of the surface measured by CloudSat's 1B-CPR algorithm. Lidar cloud mask from 2B-GEOPROF-LIDAR (Mace and Zhang, 2014) and radar cloud mask from 2B-GEOPROF (Marchand et al., 2008). The height corresponding to a range bin oscillates ± 120 m.

Conversely, CALIPSO shows a 20% increase in high clouds in the tropical tropopause compared to CloudSat due to the lidar's better detection of optically thin layers. The frequency of CALIPSO attenuation is shown in Figure 2c), indicating at least 10-15% of profiles are attenuated below 5 km and 30-70% of profiles are attenuated below 2 km globally. Cloud occurrence disappears in CloudSat below 0.5 km due to surface clutter, while CALIPSO measures cloud down to the surface if the lidar is not previously attenuated. These complementary shortcomings (e.g. CloudSat missing thin clouds, lidar missing thick clouds) can be reconciled with a merged global data product, which would sense a wider range of clouds than either instrument alone.

## 1.2 Additional value added to existing data product landscape

While many previous in-house and community datasets offer instantaneous (level 2) combined CloudSat-CALIPSO observations (e.g. Mace and Zhang (2014); Henderson et al. (2013); Sassen et al. (2008); Delanoë and Hogan (2010)), only two publicly distributed data products exist that combine CloudSat-CALIPSO observations to a global gridded dataset (Cesana, 2019; Kay and Gettelman, 2009). One of these products (Kay and Gettelman, 2009) is geared towards global clouds across the vertical column, while the other (Cesana, 2019) exclusively targets low clouds. The general-purpose product Kay and Gettelman (2009) has found wide interest in the literature for vertically-resolved climatology across the globe (e.g. Bromwich et al. (2012), Boucher et al. (2013), Houze (2014)) as well as in the study of climate processes (e.g. the sea ice–cloud feedback in Kay and Gettelman (2009)). However, both products do not extend past the 2011 CloudSat battery anomaly and transition into Daylight-Only Operations (DO-Op) mode (Nayak, 2012). While the sampling changes, including the DO-Op period more than

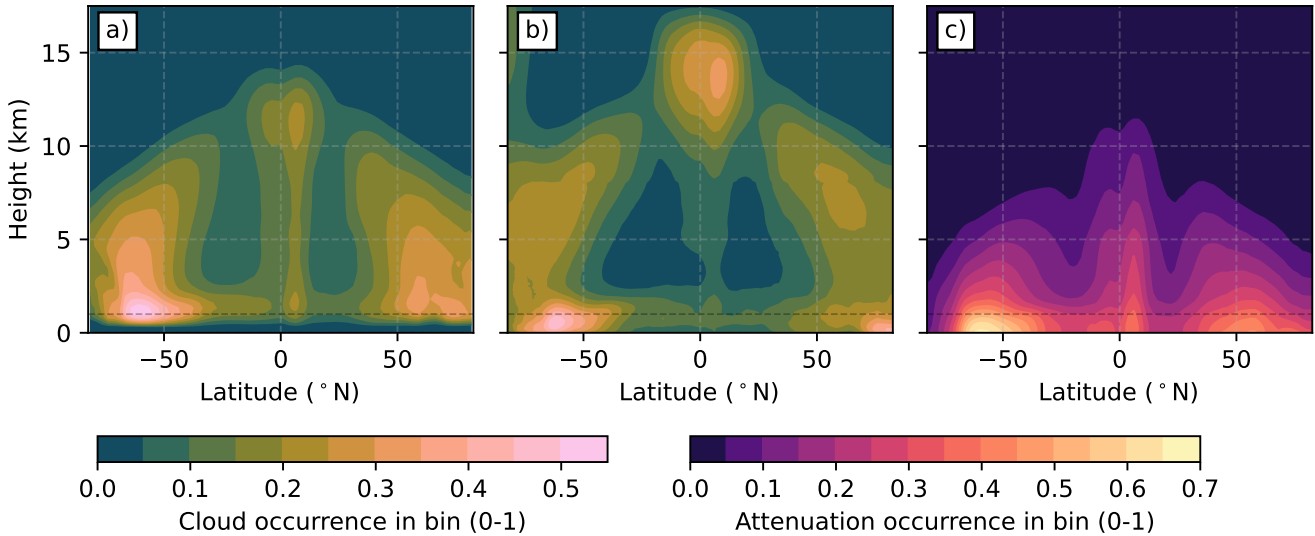

**Figure 2.** Comparison of Full-Op (2006-2011) zonal-mean vertically-resolved cloud occurrence for a) CloudSat radar (Haynes, 2020) and b) CALIPSO lidar (NASA/LARC/SD/ASDC, 2018). Panel c) shows zonal-mean frequency of lidar attenuation through the column for the same period as a) and b). Horizontal line is at 1 km altitude.

doubles the length of the data record. In the dataset presented here, we extend the record to 2020 and offer the user the choice to apply a consistent sampling methodology to the entire dataset. Since DO-Op mode decreases the already-sparse sampling

of CloudSat and CALIPSO due to their narrow swath, we also add a range of sampling variables to allow users to quantify sampling frequency. Additionally, the Kay and Gettelman (2009) dataset is calculated from an older data release (R04), which has since seen major changes, especially in CALIPSO aerosol-cloud discrimination (Mace and Zhang, 2014).

In addition to these public, global, combined CloudSat-CALIPSO datasets, authors have produced their own global data products on a per-study basis (e.g. Oreopoulos et al. (2017)), though again very few extend into the DO-Op period. In addition to

the time investment associated with the creation of personal data products, each author's approach may use different thresholds, methodologies, and processing decisions without a dedicated validation and characterization of their dataset in the literature. This especially applies to in-house products where authors use 1° grid spacing, which is problematic due to CloudSat's curtain-like swath. With the advent of in-house single-instrument level 3 data products for CloudSat and CALIPSO (Haynes, 2020; NASA/LARC/SD/ASDC, 2018, 2019), the dataset presented here can benefit from comparison to and validation against its

single-instrument counterparts, providing a presentation of the global impacts of resampling CALIPSO data to the CloudSat resolution, which has not been done previously.

Here, we present a new global, monthly data product for cloud vertical structure processed from collocated CloudSat and CALIPSO cloud mask retrievals. Our product extends the data record from 2006-2011 to 2006-2020, more than doubling the

length of the record available. It updates cloud retrievals to the latest release (R05), expands output variables (Section 3.3), and is validated against comparable single-instrument products (Section 5.1).

## 2   Input Data

Our dataset for cloud vertical structure is calculated from the level 2 datastreams 2B-GEOPROF and 2B-GEOPROF-LIDAR. The 2B-GEOPROF product offers a confidence-graded cloud mask from CloudSat's Cloud Profiling Radar (CPR), and 2B-GEOPROF-LIDAR offers the CALIPSO lidar cloud mask resampled to 2B-GEOPROF's coarser spatial and temporal grid. Both level 2 products contain time-height curtains of instrument data over an orbit, similar to Figure 1.

### 2.1   2B-GEOPROF

The 2B-GEOPROF product (Marchand et al., 2008; Marchand and Mace, 2018) contains CloudSat's hydrometeor mask. It labels regions of radar return as either surface clutter or hydrometeor and provides an estimate of the confidence of hydrometeor presence. It does not separate out cloud from precipitation, or classify hydrometeors into types.

### 2.2   2B-GEOPROF-LIDAR

The 2B-GEOPROF-LIDAR product (also called RL-GeoProf) (Mace et al., 2007; Mace and Zhang, 2014) resamples and colocates CALIPSO's native cloud mask to CloudSat's coarser vertical and temporal resolution. Since one radar volume can contain many smaller lidar volumes, the CALIPSO-only mask gives the fraction of cloudy lidar volumes contained within a radar volume. The product also contains a combined mask, which reports cloud bounds with a specific lidar and radar binary threshold applied. The input cloud mask to 2B-GEOPROF-LIDAR uses along-track averaging of up to 80 km for cloud detection (Winker et al., 2009), but 2B-GEOPROF-LIDAR only considers clouds detected using 5 km of along-track averaging.

2B-GEOPROF-LIDAR is available when CloudSat and CALIPSO footprints can be colocated to within 10 km, though footprint distance is generally less than 4 km throughout the mission (see https://www.cloudsat.cira.colostate.edu/resources/cal-cs-distance-footprints). For our product, we do not place further restrictions on footprint distance. Unfortunately, 2B-GEOPROF-LIDAR does not provide information about when the lidar is attenuated, so we estimate this information from the radar (Section 3.1) and discuss its impacts in Section 5.1. While 2B-GEOPROF-LIDAR contains a CloudSat+CALIPSO list of cloud base/-top heights based on a binary merged mask, our algorithm uses the CALIPSO-only mask in 2B-GEOPROF-LIDAR and the CloudSat-only mask in 2B-GEOPROF. This approach allows us to test the sensitivity of our results to various thresholds and calculate single-instrument auxiliary output products for validation purposes.

## 3 Methodology

3S-GEOPROF-COMB is processed in three major steps: (1) calculation of merged hydrometeor mask profiles, (2) grouping of profiles into regular grids of cells spaced at either 2.5, 5, or 10 degrees of latitude and longitude, and (3) calculation of 2D and 3D output variables summarizing the arbitrarily complex hydrometeor profiles in each grid cell.

### 3.1 Calculation of merged hydrometeor mask profiles

Our data product begins with the calculation of a binary CloudSat+CALIPSO hydrometeor mask at the orbit level 2 from 2B-GEOPROF and 2B-GEOPROF-LIDAR. We first use the 'SurfaceHeightBin' variable in 2B-GEOPROF to mask subsurface data in both 2B-GEOPROF and 2B-GEOPROF-LIDAR. This surface bin height is determined by a digital elevation model and an estimate from the surface radar return (Marchand and Mace, 2018). We also mask surface clutter in 2B-GEOPROF using the 'CloudMask' variable. We mask all profiles which have any data quality flags enabled. Then, we apply binary thresholds to the single-instrument cloud masks. For consistency with 2B-GEOPROF-LIDAR's cloud layers field, we apply a minimum threshold of 'weak echo' ($\geq 20$) to the 2B-GEOPROF cloud mask. This confidence threshold has a target false detection rate of <16% (Marchand et al., 2008). Also for consistency with 2B-GEOPROF-LIDAR, we consider a radar-lidar volume to be cloudy when at least 50% of the contained native lidar volumes are cloudy. Applying these thresholds produces a binary cloud mask for each instrument.

Prior to merging our binary cloud masks, we mask bins where lidar attenuation is likely based on the radar binary mask. If a profile transitions from radar+lidar detection of hydrometeors to radar-only detection, it is potentially attenuated. If there is no further lidar hydrometeor below the both-to-radar transition, we mask all lidar data below the both-to-radar transition. This logic is summarized in Equation (1), where $R$ ($L$) denotes radar (lidar) hydrometeor above the threshold and $i$ increases towards the surface. The result of this process can be seen in Figure 1 in profiles where the radar surface clutter was not removed. Radar surface clutter appears in the plot as missing values (solid black) above the surface height (red line). Since the lidar is masked in these profiles, it cannot be used to fill in the radar surface clutter. Without this procedure, cloud fraction <1 km would be underestimated due to the lidar filling in these bins as 'clear sky'. This technique only affects the lowest 1 km of the atmosphere.

$$\text{If there exists } k \text{ such that } \boldsymbol{X}_{RL}[i] = \begin{cases} RL & i = k-1, \\ R & i = k, \\ \neq L, RL & i > k, \text{ then mask } \boldsymbol{X}_L[i] \text{ for } i \geq k \end{cases} \tag{1}$$

After attenuated lidar is removed and binary thresholds are applied, we merge the two binary cloud masks. We consider a merged cloud mask bin to be cloudy if either radar or lidar masks are defined and above our thresholds. We consider the cloud mask to be 'clear sky' if the data is below our thresholds. If a single instrument is available (e.g. lidar in the radar surface clutter region), the combined mask is determined from that instrument alone. If neither instrument is available, for example if

the lidar is attenuated and the radar is obstructed by surface clutter, the bin is not counted for either 'cloud counts' or 'total counts' (clear-sky plus cloudy). Our cloud output variables are calculated from this merged 2D mask.

## 3.2 Auxillary single-instrument hydrometeor profiles

All following processing steps are calculated for merged hydrometeor profiles as well as single-instrument radar-only and lidar-only hydrometeor profiles. Processing of these three streams is identical except for the replacement of the merged hydrometeor profiles with single-instrument profiles. These single-instrument 3S-GEOPROF-COMB granules allow users to quantify and evaluate the relative contributions of the radar and the lidar to the merged granules. The radar-only (3S-GEOPROF-COMB-RO) and lidar-only (3S-GEOPROF-COMB-LO) granules are offered to users as separate netCDF files at the 3S-GEOPROF-COMB repository.

## 3.3 Output variables

Here we describe how 3S-GEOPROF-COMB variables are calculated from cloud mask profiles grouped into grid cells. The procedures below are repeated on each grid cell for variables with dimensions of at least latitude and longitude.

We calculate 3D cloud fraction by counting the cloudy and total number of observations at each height level. Vertically-resolved cloud counts ('cloud_counts_on_levels') reports the number of profiles with hydrometeors at each level. Vertically-resolved total counts ('total_counts_on_levels') reports the number of valid observations at each level. Cloud fraction ( 'cloud_fraction_on_levels') is the ratio of cloud counts to total counts. These variables have dimensions of latitude $\times$ longitude $\times$ height $\times$ sampling mode (Section 3.4).

While 3D cloud fraction reflects how often a height bin contains cloud, it cannot be used to infer the frequency of cloud cover over a grid cell. For this purpose, we calculate a set of 2D cloud cover variables for different types of cloud cover. 3S-GEOPROF-COMB contains high, middle, low, thick, and all cloud cover, along with unique high, middle, and low cloud cover variants. These variables report the number of profiles satisfying the following criteria:

– 'any': at least one cloud layer (thickness $\geq$ 240 m) anywhere in the profile,

– 'high': at least one cloud top above 440 mb,

– 'middle': at least one cloud with base below 440 mb and top above 680 mb,

– 'low': at least one cloud base below 680 mb,

– 'thick': at least one cloud with thickness $\geq$ 4.8 km,

– 'unique high': lowest cloud base above 440 mb,

– 'unique middle': lowest cloud base above 680 mb and highest cloud top below 440 mb,

– 'unique low': highest cloud top below 680 mb.

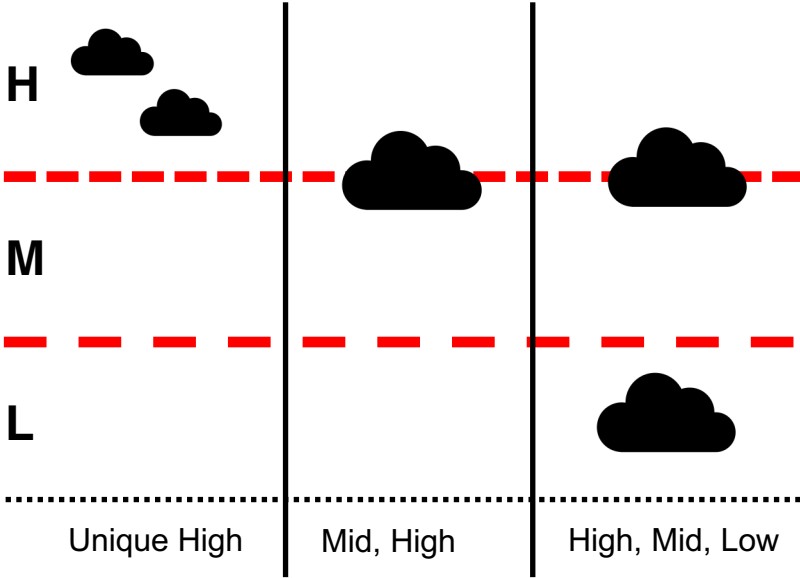

**Figure 3.** Illustration of 2D cloud cover criteria applied to schematic cloud profiles. Rows 'H', 'M', 'L' denote high-, mid-, and low-level cloud vertical regions, respectively. Text below each column indicates the 2D cloud cover criteria that each profile satisfies.

Users may select a cloud cover type via the 'type' dimension of the cloud_counts_in_column and cloud_cover_in_column variables (see Table 1). Cloud cover is the ratio of cloud counts to the total number of profiles (total_counts_in_column or total_counts_in_column_low for 'low' and 'unique low' types). We choose 680 mb and 440 mb a.m.s.l. as thresholds separating low, middle, and high cloud layers based on the International Satellite Cloud Climatology Project (ISCCP) (Rossow and Schiffer, 1999). Since our product is reported on height levels rather than pressure levels, we use the NCEP-NCAR reanalysis (Kalnay et al., 1996) to determine monthly- and zonal-mean 440 and 680 mb geometric heights for use as thresholds. The product applies 440 (680) mb height thresholds ranging from a minimum of 5.5 (2.5) at the poles to a maximum of 7 (3.5) km at the equator. These three standard layers (low-, mid-, and high-level) broadly designate clouds with different radiative feedbacks (Rossow and Schiffer, 1991; Oreopoulos et al., 2017). Note that a single profile may count for multiple categories (Fig. 3), so 'any' cloud counts will not equal the sum of the other types.

Lastly, we provide sampling information to inform users about spatiotemporal data coverage. We quantify the number of profiles ('total_counts_in_column'), the number of unique overpasses ('n_overpasses'), the number of unique days (in UTC) ('n_days'), and the statistics of the local time ('localtime_hist') that grid cells are observed. Due to high spatial correlation between profiles on a single visit to a grid cell (van de Poll et al., 2006), we recommend using the number of overpasses rather than the number of profiles when quantifying sampling. Users should be mindful that the number of overpasses required for

| Variable Name | Dimensions | Type | Description |
|---|---|---|---|
| cloud_counts_on_levels | doop, lat, lon, height | int | number of cloudy bins |
| total_counts_on_levels | doop, lat, lon, height | int | number of all bins |
| cloud_fraction_on_levels | doop, lat, lon, height | float | fraction of cloudy bins |
| cloud_counts_in_column | doop, lat, lon, type | int | number of cloudy profiles of cloud cover type |
| total_counts_in_column | doop, lat, lon | int | number of all profiles |
| total_counts_in_column_low | doop, lat, lon | int | number of all profiles for low cloud types |
| cloud_cover_in_column | doop, lat, lon, type | float | cloud cover by cloud type |
| attenuated_lidar_counts_on_levels | doop, lat, lon, height | int | number of attenuated bins |
| attenuated_lidar_counts_in_column | doop, lat, lon | int | number of profiles with attenuation |
| radar_surface_clutter_counts_on_levels | doop, lat, lon, height | int | number of radar cluttered bins |
| n_overpasses | doop, lat, lon | int | number of overpasses |
| n_days | doop, lat, lon | int | number of unique days |
| localhour22 | doop, lat, lon | int | number of profiles with local time 22:00-03:59 |
| localhour04 | doop, lat, lon | int | number of profiles with local time 04:00-09:59 |
| localhour10 | doop, lat, lon | int | number of profiles with local time 10:00-15:59 |
| localhour16 | doop, lat, lon | int | number of profiles with local time 16:00-21:59 |

**Table 1.** Data variables in 3S-GEOPROF-COMB granules. DO-Op dimension has coordinates of 'All cases' and 'DO-Op observable'. Cloud type dimension has coordinates of 'all','thick', 'high', 'middle', 'low', 'unique high', 'unique middle', and 'unique low'. See Section 3.3 for processing details.

an accurate climatology depends on a number of factors, including meteorological variability (Kotarba and Solecki, 2021; Liu, 2015; Stiller, 2010; Kotarba, 2022; Haynes, 2020). All 3S-GEOPROF-COMB data variables are listed in Table 1.

## 3.4 Treatment of Daylight-Only Operations (DO-Op) sampling

CloudSat experienced an anomaly in April 2011 which restricted the battery's capacity to charge. Fortunately, operations resumed in late 2011 but in a re-engineered Daylight-Only Operations mode (DO-Op). The anomaly and new operational mode did not change the CPR instrument, but simply restricted data collection to the sunlit portion of the orbit. The instrument powers off when it enters Earth's shadow in the Northern Hemisphere, and it powers on 9.5 minutes after leaving eclipse in the Southern Hemisphere (Witkowski et al., 2018). While this new mode results in about a 40% data loss (Kotarba and Solecki, 2021), the area in which data loss occurs varies over the course of a year (Haynes, 2020). Due to Earth's inclination, each hemisphere sees the most data loss in its respective winter and the greatest coverage in its respective summer. Since the instrument takes time to power on after entering sunlight over the southern hemisphere, the Antarctic is the region most affected by data loss.

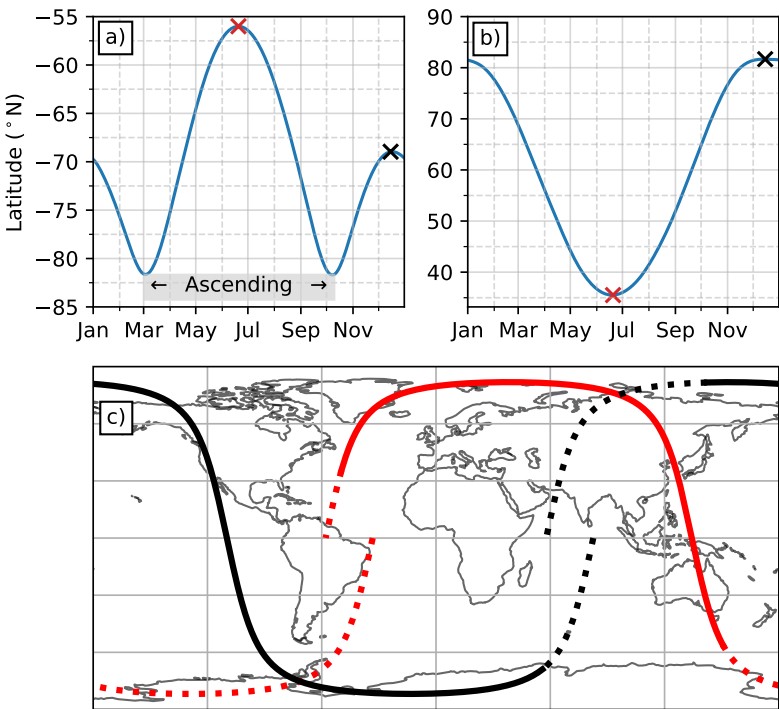

**Figure 4.** Depiction of DO-Op sampling methodology with example ground-tracks at the two extremes of the annual cycle. Panels a) and b) indicate the latitudes of the first and last DO-Op profiles, respectively. All latitudes are on the descending node of the orbit except for the portion of a) labeled 'Ascending'. Panel c) shows the DO-Op extent of two ground tracks at the extremes of the cycle, June 20th in red and December 15th in black (indicated by red and black markers panels a and b). Ground-tracks proceed in the westward direction. Solid lines indicate the DO-Op observable portion of the orbit, while dashed lines indicate the portion of the orbit not observed in DO-Op mode. Panels a) and b) are adapted from Haynes (2020).

The change from normal operations (Full-Op) to DO-Op reduces the number of observations by ∼40%, primarily restricting nighttime data on the descending branch of the orbit. We subsample Full-Op data to mimic DO-Op data collection following (Haynes, 2020; Milani and Wood, 2021) to allow users to quantify the impacts of DO-Op sampling (e.g. diurnal bias). Each orbit begins with a descending (southward) equator crossing in Earth's shadow. The satellite then enters sunlight over the Southern Hemisphere, after which the radar powers on and begins data collection. We call the latitude and branch (ascend-

ing/descending) at which this occurs the first DO-Op profile. The satellite then exits sunlight over the Northern Hemisphere and halts data collection, which we call the last DO-Op profile. Example orbits are shown in Figure 4c, where the portion of the orbit with (without) DO-Op data collection is shown as a solid (dashed) line. The portion of the orbit with DO-Op data collection varies systematically as a function of the day of year, which we leverage to implement our subsampling scheme.

        We digitize Haynes (2020)'s fitted curves indicating the latitudes of the first and last DO-Op profiles as a function of day of

210    year, shown in Figure 4a) and b). The first DO-Op observable profile (Fig. 4a) is located on the ascending (northward) branch

of the orbit March through August, and on the descending branch otherwise. The last DO-Op observable profile (Fig. 4b) is located on the descending branch year-round. Example ground-tracks at the extremes of this annual cycle are shown in Figure 4c), with the DO-Op observable portion of the orbits shown as solid lines.

3S-GEOPROF-COMB is computed with and without Full-Op subsampling to DO-Op collection patterns. Users are given the choice to apply this subsampling via the 'doop' dimension. The coordinate 'DO-Op observable' gives the data product computed using only profiles that either were or would have been collected in DO-Op mode. The coordinate 'All cases' gives the data product computed with all observations with no subsampling applied. After the start of DO-Op mode, these two coordinates give the same values since no subsampling is applied. This subsampling option allows users to test the effects of DO-Op mode on their analyses or apply a consistent sampling pattern to the entire dataset (e.g. for trends or interannual variability).

## 4 Output Files

3S-GEOPROF-COMB offers globally gridded, temporally aggregated files containing the cloud and sampling data variables described in Section 3.3. Output files are processed at monthly and seasonal timescales, and at $2.5° \times 2.5°$, $5° \times 5°$, and $10° \times 10°$ longitude by latitude spatial scales. Vertically-resolved cloud occurrence has dimensions of DO-Op (Sec. 3.4), latitude, longitude, and height. Vertically-integrated cloud cover has dimensions of DO-Op, latitude, longitude, and simplified cloud type. All cloud variables are reported as raw counts and occurrence fractions. All output variables are listed in Table 1. Since counts are given, users may weight data according to their own spatial and temporal aggregations. Output files are available for combined radar+lidar, radar-only, and lidar-only cloud fields, with otherwise identical processing.

### 4.1 Data Coverage

3S-GEOPROF-COMB is only processed when both 2B-GEOPROF and 2B-GEOPROF-LIDAR are available and less than 50% of data is missing. For example, monthly files would require 14 days worth of data, 6 weeks of data for seasonal files, etc. Figure 5 shows the number of input granules available per month for our data streams along with the total duration (in days) of observations. Since 2B-GEOPROF-LIDAR is only available when 2B-GEOPROF is also available, the line for 2B-GEOPROF-LIDAR in Figure 5 indicates the number of input granules used in our data product. Our requirement of 50% data availability is not a threshold for accurate climatologies, since this depends on the requirements of the study and the meteorological variability in the region(s) of interest (e.g. Kotarba and Solecki (2021)).

While data are available 2006-2020, two prolonged data outages have occurred. The first outage (April 2011 to May 2012) was caused by the CloudSat battery anomaly, and the second outage (January to October 2018) was caused by a CloudSat reaction wheel anomaly. In both cases, CloudSat leaves formation flying and the return of 2B-GEOPROF-LIDAR is delayed as CloudSat waits to rejoin CALIPSO. After the 2011 anomaly, CloudSat rejoined the A-train in DO-Op mode. After the May 2018 anomaly, CloudSat left the A-train and was joined by CALIPSO on a secondary orbit called the C-train in October 2018. CloudSat suffered another reaction wheel anomaly in August 2020, after which instrument pointing accuracy was degraded,

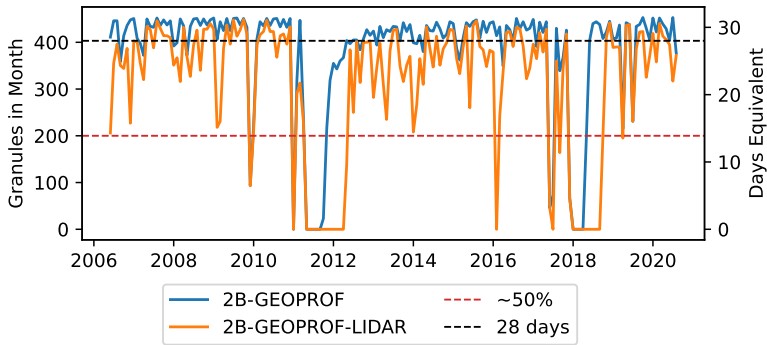

**Figure 5.** Input data availability for monthly files over the course of the mission. 2B-GEOPROF-LIDAR is only processed when 2B-GEOPROF is available. Horizontal lines correspond to near-complete data availability (403 granules ∼ 28 full days, black line) and the threshold below which monthly output files are not processed (200 granules, red line). Outage in April 2011 corresponds to the CloudSat battery anomaly. 2B-GEOPROF-LIDAR returns when CloudSat rejoined the A-train in DO-Op mode on 15 May 2012. Outage in May 2018 corresponds to a CloudSat reaction wheel anomaly. 2B-GEOPROF-LIDAR returns when CALIPSO exits the A-train to join CloudSat. Outage in August 2020 corresponds to a further reaction wheel anomaly in CloudSat. Data collection resumed in December 2021 with ACT-TWO DOOP mode, but 2B-GEOPROF-LIDAR is unlikely to return due to variable instrument pointing.

complicating future prospects of colocating with CALIPSO. Figure 5 represents offerings at the CloudSat Data Processing Center (DPC) (cloudsat.cira.colostate.edu). 3S-GEOPROF-COMB will be updated as new input data becomes available, which will likely extend the record up to August 2020.

## 4.2 Example plots

Some example plots of 3S-GEOPROF-COMB are shown in Figure 6. Figure 6a shows 'all' cloud cover over the full dataset (2006-2020) at 2.5° resolution, which has been shown to agree with MODIS by Mace and Zhang (2014). Figure 6b shows 2006-2020 zonal-mean cloud fraction at 2.5° resolution. Obviously and impressively, the zonal mean cloud fraction structure is consistent with global zonal mean atmospheric circulation. Ascending regions have high cloud fraction throughout the troposphere, while subsiding regions have cloud only in the lowest 2.5 km of the atmosphere. Additionally, the lowest cloud fraction contour (0.05) shows the poleward decrease of the tropopause height.

The combined zonal-mean cloud fraction (Fig. 6b) matches CALIPSO where we expect the lidar to perform better than the radar. For example, the equatorial cirrus plume from 10-16 km resembles CALIPSO cloud fraction (Fig. 2b) in shape and magnitude much more strongly than CloudSat cloud fraction (Fig. 2a). Conversely, 3S-GEOPROF-COMB matches CloudSat in regions with frequent lidar attenuation. For example, equatorial deep convection (3-6 km) matches CloudSat's 20% cloud fraction rather than CALIPSO's 5%. In the polar and extra-tropical latitudes, the combined product tracks CloudSat's smooth decrease in cloud fraction with height while preserving CALIPSO's higher cloud fraction for near-surface clouds (< 1 km).

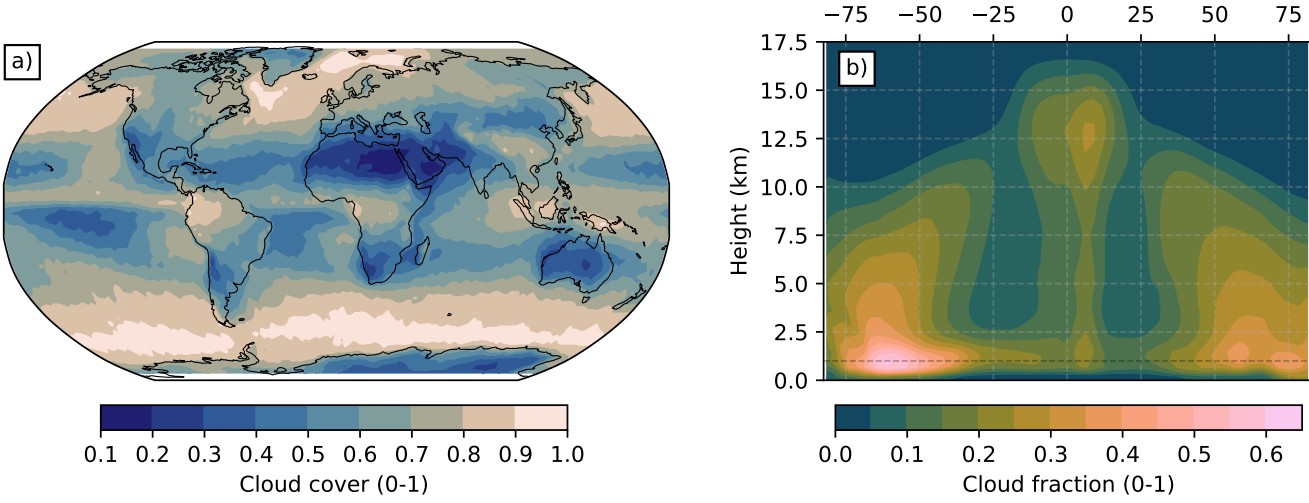

**Figure 6.** 3S-GEOPROF-COMB 2006-2020 (a) annual mean all cloud cover and (b) zonally-averaged cloud fraction. No DO-Op subsampling is applied to the Full-Op period.

Overall, 3S-GEOPROF-COMB combines the detection strengths of the two instruments for a more complete measurement of global clouds.

### 4.3 Sampling characteristics

Data users must be mindful of CloudSat+CALIPSO's narrow transect sampling. This sampling can become sparse with missing input data and must be balanced by an appropriate choice of spatial and temporal resolution. The impacts of transect sampling on climatology uncertainty has been studied in general (Liu, 2015; Stiller, 2010; van de Poll et al., 2006) and specifically in the context of CloudSat+CALIPSO Full-Op data (Kotarba and Solecki, 2021; Kotarba, 2022). Fewer studies have investigated DO-Op sampling (Milani and Wood, 2021). Kotarba and Solecki (2021) found that regional variations in cloud variability are the largest source of CloudSat+CALIPSO vertical cloud fraction uncertainty. This cloud variability uncertainty is greater than the influence of the choice of spatial or temporal resolution. To reduce this uncertainty, we encourage users to balance coarse spatial resolution with fine temporal resolution, and vice versa, as well as to consider the cloud variability in their geographical and height levels of interest.

Major month-to-month variations in DO-Op coverage can bias multi-month averages without proper weighting from users. Figure 7 shows the number of overpasses as a function of latitude for July (first row, a-c) and December (second row, d-f) of 2010, given by the 'n_overpasses' variable in 3S-GEOPROF-COMB. These two months lie at the extremes of the seasonal cycle of DO-Op sampling (Fig. 4). Above 45°N, July has no reduction in sampling from Full-Op to DO-Op (Fig. 7c), regardless of grid size, whereas December half as many observations in DO-Op compared to Full-Op. Since more DO-Op observations are taken in warmer months in the North Hemisphere, yearly averages without accounting for these variations would preferentially

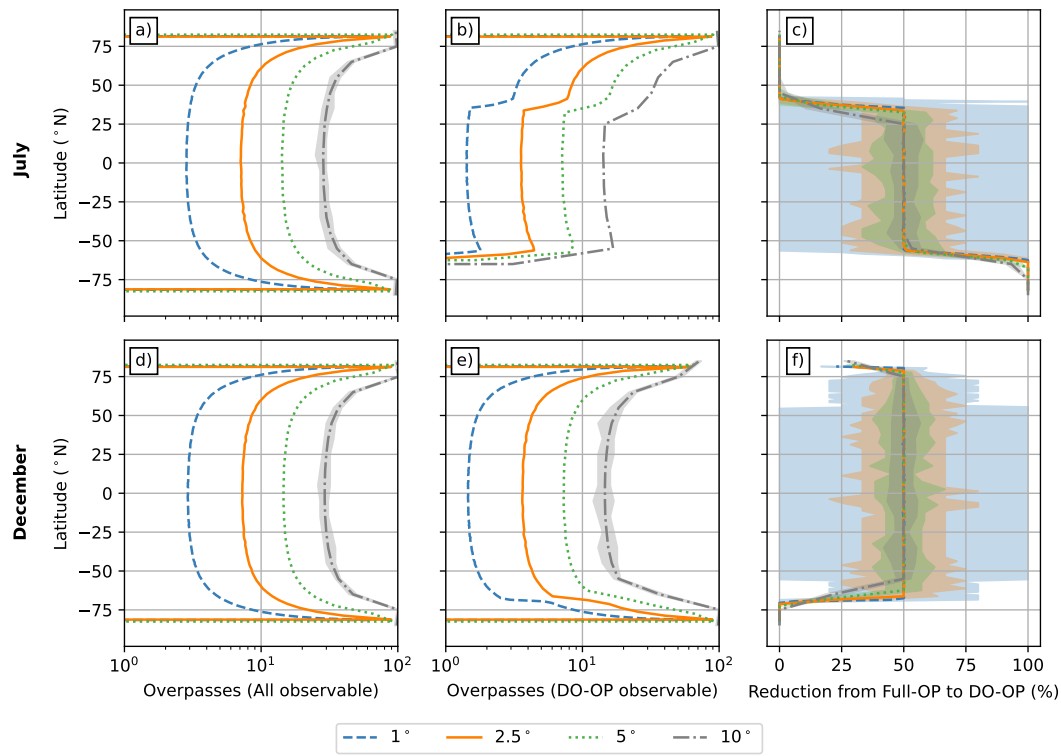

**Figure 7.** Example overpass statistics for July 2010 (a-c) and December 2010 (d-f) as a function of latitude. Lines show zonal-mean over-passes while shaded areas show the range of zonal variations for 1, 2.5, 5, and 10° resolution. Left column (a,d) shows Full-Op sampling, middle column (b,e) shows DO-Op subsampling, and right column (c,f) shows the percent reduction in overpasses from Full-Op to DO-Op.

weight JJA. Users can avoid this issue by weighting each month by the number of profiles ('total_counts_in_column') or the number of overpasses ('n_overpasses') when averaging over different months. 3S-GEOPROF-COMB seasonal output files report month-unweighted cloud variables.

Smaller grid sizes (e.g. 1°, 2.5°) both reduce the frequency of overpasses and introduce zonal variations in sampling. If users have specific sampling/significance needs for a region of interest, these zonal fluctuations may be undesirable. The extent of fluctuations is shown in Figure 7, where zonal-mean overpasses are indicated by lines and the range of zonal variation is indicated by shaded areas. The finest resolution, 1°, ranges from 0 to 4 (0 to 2) overpasses per month in Full-Op (DO-Op) outside of polar regions, i.e., some grid cells are never observed. With the reduction from Full-Op to DO-Op (Fig. 7c,f),

some grid cells are totally removed (100% reduction) while others are unaffected (0% reduction), indicating that DO-Op introduces further spatial heterogeneity to the sampling at this fine resolution. For these reasons, we do not distribute 1° files in 3S-GEOPROF-COMB and choose 2.5° as the minimum acceptable resolution, though coarser grids (e.g. 10°, grey shaded area Fig. 7) further mitigate these effects. This same reasoning applies at seasonal and yearly temporal resolution, since CloudSat+CALIPSO ground tracks repeat every 16 days.

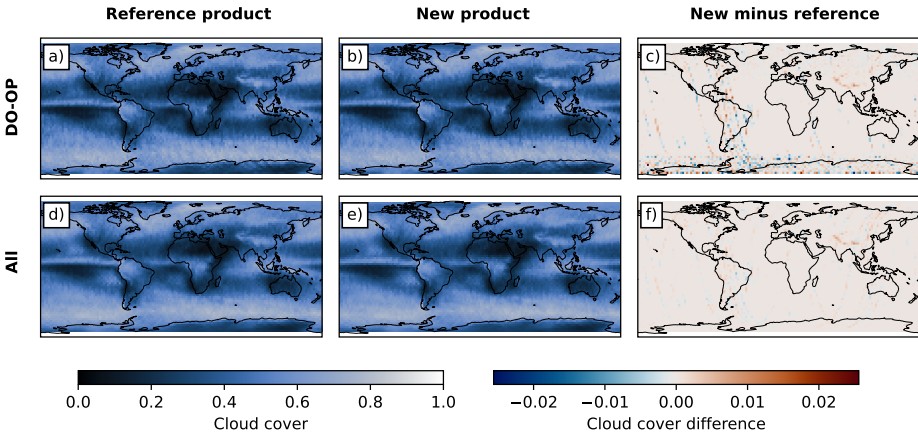

**Figure 8.** Comparison between our 3S-GEOPROF-COMB radar-only ('3GC-RO2') (b,e) and the in-house CloudSat-only product 3S-RMCP (Haynes, 2020) (a,b) cloud cover for DO-Op observable (a-c) and all observations (d-f) for the Full-Op (2006-2011) period. Third column (c,f) shows the mean of all 3GC-RO2 minus 3S-RMCP monthly differences.

Users should also note that due to surface clutter, radar observations are reduced from 500-1000 m and nearly eliminated between 0-500 m above ground level. Cloud fraction is not reliable when both radar clutter ( 'radar_surface_clutter_counts_on_levels' ) and lidar attenuation ( 'attenuated_lidar_counts_on_levels' ) are frequent compared to the number of valid observations ( 'total_counts_on_levels'). In regions of high elevation, users should consult 'radar_surface_clutter_counts_on_levels' to identify the heights at which cloud fraction may be unreliable.

## 5  Validation

### 5.1  Comparison to Level 3 CloudSat-only and CALIPSO-only products

By comparing our dataset to CloudSat-only and CALIPSO-only data products (Haynes, 2020; NASA/LARC/SD/ASDC, 2018, 2019) we can validate our processing methodology and verify the added value of the combined product. For this comparison, we use variants of 3S-GEOPROF-COMB processed from single-instrument, rather than combined, hydrometeor profiles. For lidar validation, we use the standard lidar-only version of our product 3S-GEOPROF-COMB-LO (Sec. 3.2), which we call '3GC-LO' here. For radar validation, we processes a radar-only variant including all 2B-GEOPROF granules instead of only those for which 2B-GEOPROF-LIDAR is also present (Sec. 4.1) for consistency with the radar product 3S-RMCP (Haynes, 2020). We designate this expanded radar-only variant as 3S-GEOPROF-COMB-RO2, which we call '3GC-RO2' here. Note that for consistency with 3S-RMCP, we include radar surface clutter counts under total observations for this comparison.

For the radar, we compare 3GC-RO2 to 3S-RMCP cloud cover and zonal-mean cloud fraction at $2.5° \times 240$ m for all months for which both products are available (2006-2016). Cloud cover for all observations (Fig. 8a-b) and DO-Op subsampled observations (Fig. 8d-e) are qualitatively indistinguishable, with the mean difference between individual months not exceeding

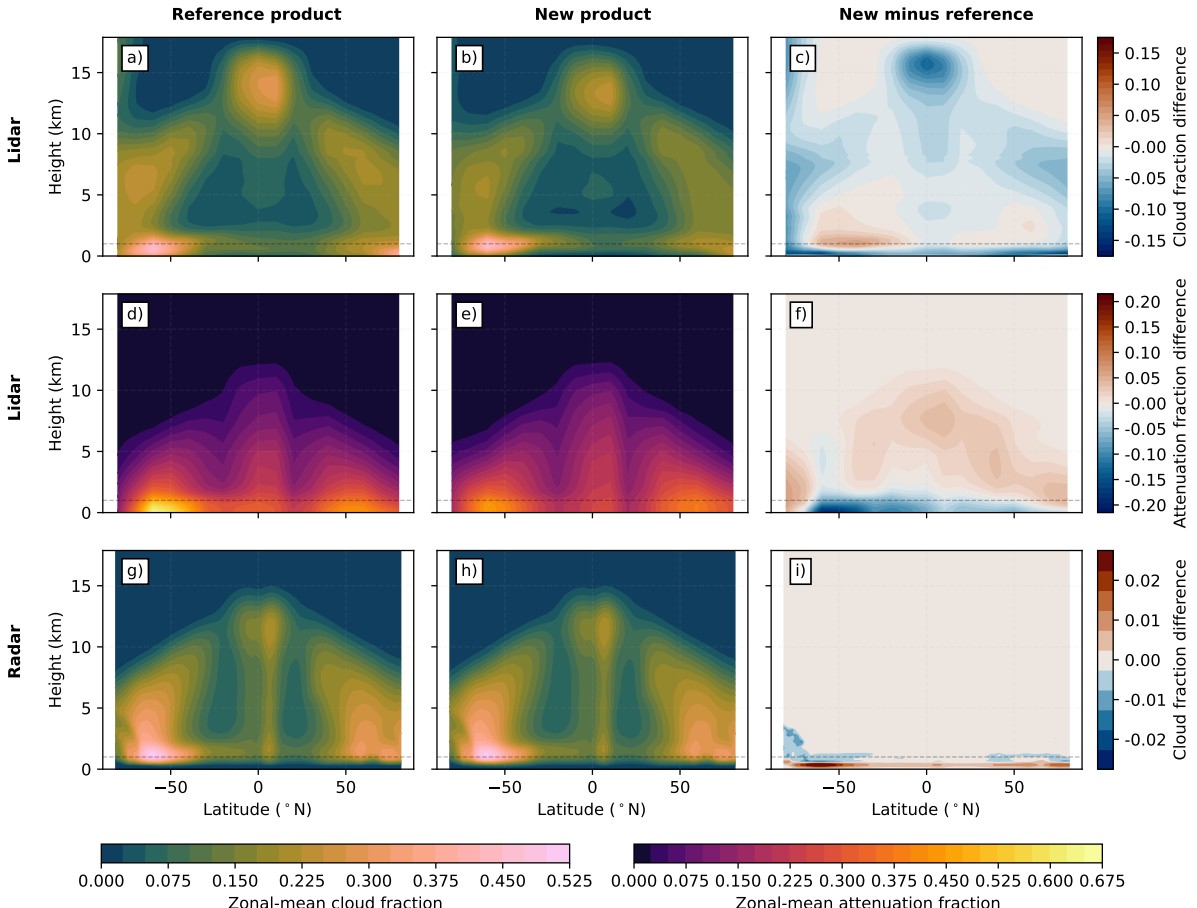

**Figure 9.** Zonal-mean cloud fraction and attenuation fraction comparison between single-instrument 3S-GEOPROF-COMB and pre-existing single-instrument level 3 datasets. Panels a-c show 3S-GEOPROF-COMB lidar-only ('3GC-LO') (a), CALIPSO Cloud Occurrence ('CAL-COS') (b), and 3GC-LO minus CAL-COS cloud fraction. Panels d-f show CC-LO (d), CAL-CF (e), and CC-LO minus CAL-COS (f) attenuation fraction. Panels g-i show 3S-GEOPROF-COMB radar-only ('CC-RO2') (g), the CloudSat-only 3S-RMCP (h), and 3GC-RO2 minus 3S-RMCP (i) cloud fraction. Contours for lidar difference plots (c,f) are spaced at 0.01, radar difference plot contours are spaced at 0.005 (i). Contour spacing for all other plots is 0.025. Lidar comparison (a-f) is for the Full-Op period, while radar comparison is for the full record (Full-Op and DO-Op). CAL-COS dataset is resampled to 10° horizontal grid and 240 m vertical grid to coincide with 3GC-LO.

0.026. Non-zero differences occur along individual ground-tracks, suggesting that discrepancies are due to minor differences in the input granules available from the CloudSat Data Processing Center when each product was produced. In terms of zonal-mean cloud fraction, 3GC-RO2 is identical to 3S-RMCP (Fig. 9g-i) above the surface, with slight (< 0.03) differences within 1 km of the surface. These minor near-surface differences are likely due to interpolation error when accounting for the difference in height levels between the two products, since 3GC-RO2 was linearly interpolated to 3S-RMCP's height levels

for comparison. From this strong agreement, we conclude that 3S-GEOPROF-COMB successfully replicates 3S-RMCP cloud fields and DO-Op subsampling.

For the lidar, we compare 3GC-LO to CALIPSO Cloud Occurrence Standard ('CAL-COS') (NASA/LARC/SD/ASDC, 2018) zonal-mean cloud fraction (Fig. 9a-c). We do not compare 3GC-LO to CALIPSO cloud cover (NASA/LARC/SD/ASDC, 2019) since this product is calculated using a different definition of cloud cover from the one used here. We coarsen CAL-COS to 10° horizontal and 240 m vertical resolution to align the two datasets' spatial grids.

Several differences are present due to differences in the underlying retrievals. Our 3GC-LO shows decreased high clouds
compared to CAL-COS, reaching a peak 0.11 decrease at 15.7 km altitude over the equator. This difference is likely due to thin cirrus only detected at 20 km and 80 km along-track averaging lengths, which are excluded by the input product used in 3GC-LO (Sec. 2.2, Mace and Zhang (2014)). 3GC-LO also shows decreased very near-surface (< 500 m) cloud compared to CAL-COS, reaching up to a 0.16 reduction at 120 m altitude in the polar latitudes. This decrease is likely due to underestimated attenuation (discussed below) and the coarsening of CALIPSO profiles to 240 m vertical resolution, which removes clouds
with thickness < 120 m. Low cloud fraction 0.5-3 km is increased up to 0.05, primarily over the Southern Ocean. Low cloud fraction 0.5-3 km increases by up to 0.05, primarily over the Southern Ocean. This increase is due to the fact that CAL-COS excludes shallow marine liquid clouds detected using along-track averaging, which tends to overestimate cloud cover (NASA/LARC/SD/ASDC, 2019, Detailed Data Quality Summary).

Additionally, we compare our zonal-mean estimated lidar attenuation fraction (Sec. 3.1) to the actual lidar attenuation given
in CAL-COS. Overall, 3S-GEOPROF-COMB agrees with CAL-COS (Fig. 9d-f), where 3S-GEOPROF-COMB attenuation fraction is between 0.1 greater than and 0.2 less than CAL-COS, with the greatest differences in near-surface polar regions (Fig. 9f). The decreased attenuation over the Southern Ocean compared to the increased attenuation over the Arctic suggests that our algorithm for estimating attenuation (Sec. 3.1) is sensitive to the prevailing cloud and precipitation regime. In particular, some attenuation differences may arise from warm marine clouds which are opaque to CALIPSO but go undetected
by CloudSat (Liu et al., 2016, 2018), which Liu et al. (2018) found to be globally most prevalent over the Southern Ocean. These differences would impact cloud fraction by increasing or decreasing the number of total (clear-sky+cloudy) observations. Increased attenuation would increase cloud fraction by decreasing the number of total observations, and vice versa. While this could explain the < 500 m reduced cloud fraction noted above, it does not explain the increased cloud fraction over the Southern Ocean, since decreased attenuation would decrease cloud fraction. Additionally, increased attenuation 3-10
km is associated with decreased cloud fraction, so attenuation does not explain the reduction. The general lack of correlation between attenuation and cloud fraction differences further suggests that discrepancies are driven by differences between 2B-GEOPROF-LIDAR and the native CALIPSO cloud retrievals. We note the overall good agreement between these products when these differences are considered.

## 5.2 Comparison to ground-based sites

We compare 3S-GEOPROF-COMB to four Atmospheric Radiation Measurement (ARM) ground sites, which offer cloud mask retrievals from a combination of ground-based radar and lidar (Xie et al., 2010). We choose ARM sites in Graciosa Island,

Azores, Portugal in the Eastern North Atlantic (ENA C1); Utqiagvik, Alaska on the North Slope of Alaska (NSA C1, Verlinde et al. (2016)); Lamont, Oklahoma in the Southern Great Plains (SGP C1, Sisterson et al. (2016)); and Darwin, Australia in the Tropical Western Pacific (TWP C3, Long et al. (2016)). These sites represent a wide variety of cloud regimes for evaluation

of our satellite-derived cloud product. Many studies have already compared surface-based radar/lidar and CloudSat/CALIPSO cloud measurements (e.g. Liu et al. (2010); Blanchard et al. (2014); Liu et al. (2017); Protat et al. (2014); Kim et al. (2008)), to which we refer the reader for more in-depth discussion. Our intent is to broadly compare multi-year climatologies from 3S-GEOPROF-COMB to explore the utility of our product.

We compare surface-derived cloud cover and vertical cloud fraction to $2.5° \times 2.5°$ and $5° \times 5°$ 3S-GEOPROF-COMB grid

cells containing each ARM site (Fig. 10). Averages are calculated from all months where both surface and satellite observations are available (see Figure 5). The SGP and NSA comparisons are calculated from ∼11 years of data, while the TWP and ENA comparisons are calculated from ∼4 years of data (2006-2009 and 2015-2020, respectively). 3S-GEOPROF-COMB broadly captures the shape and magnitude of each location's seasonal cycle (Fig. 10, first row), where the greatest differences are decreased satellite cloud cover in regions with more low clouds (ENA, Fig. 10a and NSA, Fig. 10d). If we exclude clouds

with top heights below 500 m from the surface-based measurements, the region where surface clutter prevents CloudSat measurements, the agreement improves (Fig. 10, first row, dash-dotted line). However, by excluding the region where surface clutter reduces CloudSat sensitivity (500-1000 m), the surface-derived cloud cover drops below satellite-derived at NSA (Fig. 10d) (for other sites >500 m cover equals >1 km cover), suggesting that 3S-GEOPROF-COMB still contains relevant cloud information at altitudes 500-1000 m.

We also compare vertical cloud fraction for the month of best (Fig. 10, middle row) and worst (Fig. 10, last row) cloud cover agreement at each site. 3S-GEOPROF-COMB generally shows more high clouds than the surface-based measurements, likely due to the difficulty of capturing optically thin clouds far from the surface (Kim et al., 2008; Protat et al., 2014). Our product broadly captures the vertical cloud fraction for each month, though several comparisons have large ($\geq 0.1$) differences. The largest occurs in April at the TWP site (Fig. 10l), where satellite-derived cloud fraction is uniformly much larger than

surface-derived cloud fraction. This difference likely comes from radar attenuation from the frequent heavy precipitation at the site (Long et al., 2016; Liu et al., 2010), where Liu et al. (2010) noted good ARM-CloudSat cloud fraction agreement for non-precipitating cases. Other points of note are January at ENA (Fig. 10a) and October at NSA (Fig. 10e), where the product does a good job of estimating cloud cover but overestimates the <1 km low cloud fraction peak. This is likely because of the finer vertical resolution (40 vs 240 m) of the surface-based instruments compared to 3S-GEOPROF-COMB, where the surface-

based instruments spread the overall cloud occurrence across a wider array of height bins, thereby decreasing maximum cloud fraction.

## 6   Conclusions

In this paper, we document our efforts to combine observations from spaceborne radar (CloudSat) and lidar (CALIPSO) to make a new global gridded product of monthly cloud vertical fraction and cloud cover. Building on previous efforts, our level-3 prod-

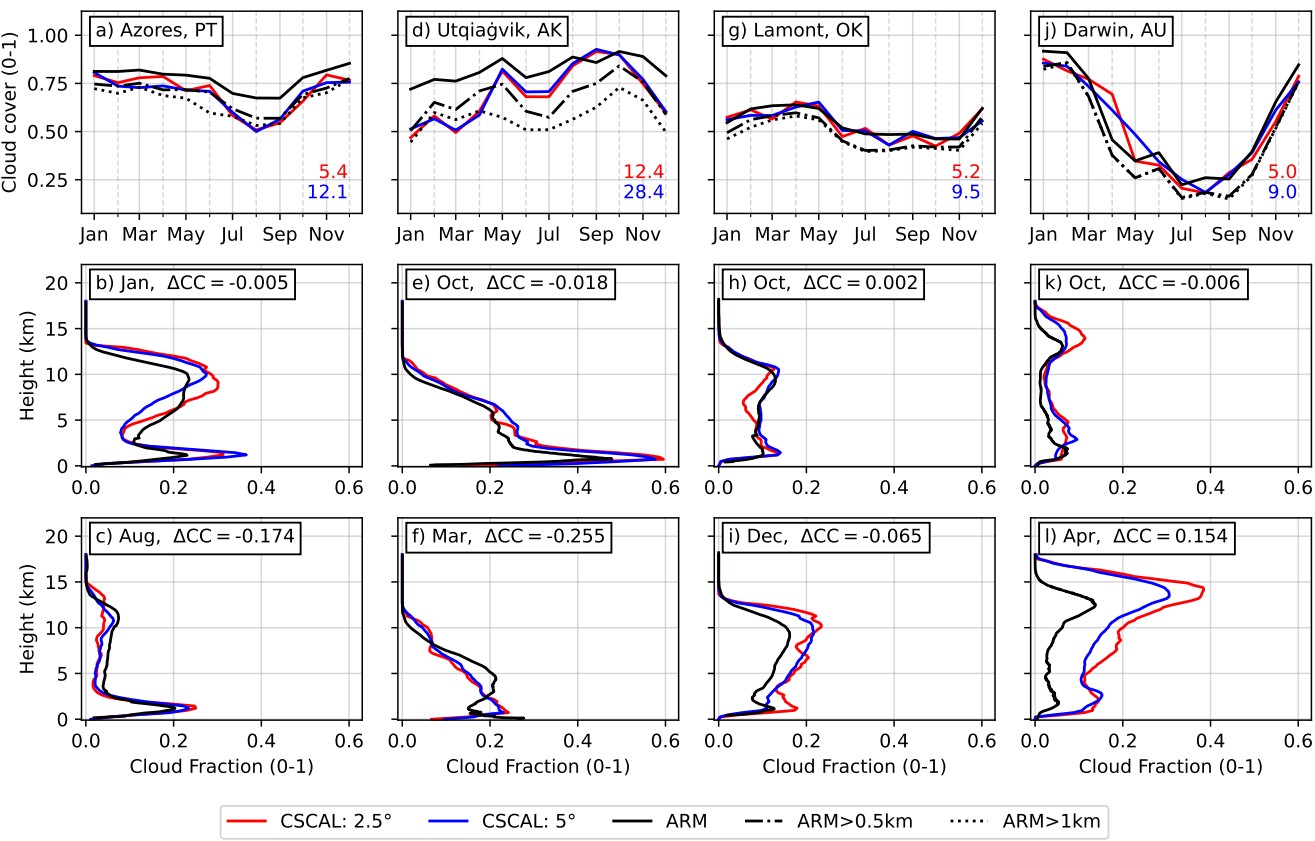

**Figure 10.** Comparison between 3S-GEOPROF-COMB and Atmospheric Radiation Measurement (ARM) ground sites at Graciosa Island, Azores, Portugal 2015-2020 (a-c); Utqiagvik, Alaska 2006-2020; Lamont, OK 2006-2020 (d-f); and Darwin, Australia 2006-2009 (j-l). Ground site cloud retrieved from a combination of cloud radar, micropulse lidar, and ceilometer. The first row shows satellite-derived (colored lines) compared to surface-derived (black lines) cloud cover across sites, the middle row shows vertical cloud fraction for month of best cloud cover agreement, and the last row shows vertical cloud fraction for month of worst cloud cover agreement. Surface-based cloud cover is shown for all clouds (solid line), only clouds > 500 m (dash-dotted line), and only clouds > 1000 m (dotted line). Colored text in the first row indicates the average number of overpasses per month at 2.5° (red) and 5° (blue) grid cells. Text ('Δ CC') in the bottom two rows indicates the difference between 5° satellite and surface total cloud cover. Satellite information is shown for product 2.5° × 2.5° and 5° × 5° grid cells containing the ARM sites.

uct called 3S-GEOPROF-COMB combines existing level-2 CloudSat data products (2B-GEOPROF, 2B-GEOPROF-LIDAR) over the entire globe for the full available observing period (2006-2020) from the latest release (R05). Full documentation of methods and data included are provided in this paper, and the data are publicly available for all to use at a Zenodo repository https://doi.org/10.5281/zenodo.8057790 (Bertrand et al., 2023). After peer review of the dataset, the product will be migrated to long-term hosting at the NASA Atmospheric Science Data Center (ASDC) Distributed Active Archive Center (DAAC).

We anticipate use by the scientific community especially for studying cloud processes and cloud-climate-circulation coupling. While quantitative comparison of cloud amount in observations and models should use a satellite simulator (Bodas-Salcedo et al. 2011, Kay et al. 2012), qualitative model evaluation can be done using our globally gridded product. We also anticipate this dataset as a candidate baseline climate data record to be compared with future active cloud remote sensing missions including combined spaceborne radar and lidar. Future missions that could benefit from comparison with our product include Earth-

CARE (Illingworth et al. 2015) and Atmosphere Observing System (AOS, https://aos.gsfc.nasa.gov/). Scheduled for launch in 2024 as a joint ESA (European Space Agency)/JAXA (Japan Aerospace Exploration Agency) mission, EarthCARE includes spaceborne radar and lidar. Scheduled to launch in the late 2020s and supported by multiple space agencies including National Aeronautics and Space Administration (NASA), Japan Aerospace Exploration Agency (JAXA), National Centre for Space Studies (CNES), Canadian Space Agency (CSA) and German Aerospace Center (DLR), AOS includes spaceborne radars,

lidars, polarimeter, microwave radiometer, and far-infrared imaging radiometer.

## 7 Code and data availability

3S-GEOPROF-COMB, along with the single-instrument variants, are available to users at the Zenodo repository https://doi.org/10.5281/zenodo.8057790 (Bertrand et al., 2023). The code used to produce 3S-GEOPROF-COMB and examples of how to use the product are available at https://github.com/bertrandclim/3S-GEOPROF-COMB. The data and code used to produce the

figures in this paper are available at https://github.com/bertrandclim/essd2023. The satellite datasets used for the production and validation of 3S-GEOPROF-COMB are available at the CloudSat Data Processing Center (DPC) and the Atmospheric Science Data Center Distributed Active Archive Center (ASDC DAAC). The ground-based dataset used for validation, ARM-BECLDRAD, is available from the Atmospheric Radiation Measurement (ARM) Data Center (ADC).

*Author contributions.* WB produced the data product with input and guidance from JK and JH. WB prepared the manuscript with contribu-
tions from all co-authors.

*Competing interests.* The authors declare that no competing interests are present.

*Acknowledgements.* This work was supported by NASA CloudSat/CALIPSO Science Team grant 80NSSC20K0135 (WB, JEK) and by the US Department of Energy (DOE) Atmospheric System Research (ASR) program grant DE-SC0013306 (WB). This work utilized the Alpine high performance computing resource at the University of Colorado Boulder. Alpine is jointly funded by the University of Colorado Boulder,
the University of Colorado Anschutz, Colorado State University, and the National Science Foundation (award 2201538). The authors thank David Winker and one anonymous reviewer for their insightful comments on the manuscript.

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
