# Peer review of "A Global Gridded Dataset for Cloud Vertical Structure from Combined CloudSat and CALIPSO Observations"

_Earth System Science Data, 2023_

## Author Comment (AC1)

**Response to Reviewers**

The authors thank the reviewers for their time and comments on the manuscript. Our responses and changes (italics) are provided below each reviewer comment (bold).

To summarize, the main changes made are:
1. The data product now uses 440 mb and 680 mb geometric heights for cloud cover type that vary as a function of latitude and month (Reviewer 2).
2. The comparison to ground-based sites has been expanded from 1 to 4 sites (Reviewer 1).
3. Since submission of the paper, additional input data (July 2019 – August 2020) have been released, which we have included in our product and revised manuscript.
4. Discussions of uncertainty (Reviewer 1), Daylight-Only Operations sampling (Reviewer 2), and instrument capabilities (Reviewer 2) have been expanded in response to reviewers.

**Response to Reviewer #1**

**The manuscript delivers a comprehensive exploration into the creation of a global dataset that delineates the vertical structure of clouds, harnessing the combined observational capabilities of CloudSat and CALIPSO. The attempt to integrate data from these two satellites is both ambitious and timely, providing a unified perspective on cloud distribution, altitude, and overall cloud architecture. The result is a dataset that could become an invaluable tool for climatologists, meteorologists, and researchers working in atmospheric sciences.**

We thank reviewer #1 for their positive review and constructive comments that have improved the manuscript.

**In Figures 6 and 9, the portrayal of global cloud patterns and anomalies is instructive. However, it would enhance the paper's depth to delve into more elaborations about regional variations, the role of seasons, and potential correlations with established climatic phenomena. In addition, the authors may discuss whether the datasets can potentially illuminate some new insights into cloud characteristics.**

We agree with the reviewer #1. Much can be learned from a more detailed analysis of this dataset. Indeed, we look forward to future studies using these data. As this paper is for the data journal ESSD, this paper focuses on data product design, not data analysis. Detailed analysis of regional variations, the role of seasons, and correlations with climate variables is beyond the scope of this current paper.

That said, we agree it is useful to relate the underlying process to our figures when the connections are impressive and obvious. As such, we have added the following text:

*"Figure 6b shows 2006-2020 zonal-mean cloud fraction at 2.5° resolution. Obviously and impressively, the zonal mean cloud fraction structure is consistent with global zonal mean atmospheric circulation. Ascending regions have high cloud fraction throughout the troposphere, while subsiding regions have cloud only in the lowest 2.5 km of the atmosphere. Additionally, the lowest cloud fraction contour (0.05) shows the poleward decrease of the tropopause height."* (see revised lines 247-250)

**It's laudable that the authors have illuminated potential discrepancies and constraints within the dataset in the error analysis section. Nonetheless, a more expansive discussion on the inherent uncertainties, especially when navigating regions marked by different topographies or areas known for complex cloud systems, would be invaluable.**

We thank reviewer #1 for raising an important but challenging topic for any data paper: uncertainty. We have discussed and addressed uncertainty in our paper through comparison to existing data products, and by outlining the impacts of our decisions on cloud detection. We also encourage the reviewer and users of our dataset to read the quality assessment of our input data products and measurements (Marchand et al. 2008; Mace and Zhang 2014; Winker et al. 2007).

The largest uncertainty in our product is the sampling uncertainty, i.e., the uncertainty of the spatiotemporal means reported. We discuss this in section 4.3, "Sampling Characteristics". Because it is particularly relevant and useful to the topic of uncertainty, we added discussion of a paper that found that cloud variability has a major influence on sampling uncertainty: Kotarba and Solecki (2021). See lines 265-269 of the revised paper:

*"Kotarba and Solecki (2021) found that regional variations in cloud variability are the largest source of CloudSat+CALIPSO vertical cloud fraction uncertainty. This cloud variability uncertainty is greater than the influence of the choice of spatial or temporal resolution. To reduce this uncertainty, we encourage users to balance coarse spatial resolution with fine temporal resolution, and vice versa, as well as to consider the cloud variability in their geographical and height levels of interest."*

We agree it is important to mention uncertainties related to topography. These uncertainties are already addressed through data flags in the input data products we use (Marchand and Mace 2018, section 4.1.2). As such, we have added the following paragraph at lines 289-293:

*"Users should also note that due to surface clutter, radar observations are reduced from 500-1000 m and nearly eliminated between 0-500 m above ground level. Cloud fraction is not reliable when both radar clutter (* radar_surface_clutter_counts_on_levels *) and lidar attenuation (* attenuated_lidar_counts_on_levels *) are frequent compared to the number of valid observations (* total_counts_on_levels *). In regions of high elevation, users should consult* radar_surface_clutter_counts_on_levels *to identify the heights at which cloud fraction may be unreliable."*

Kotarba, A. Z. and Solecki, M.: Uncertainty Assessment of the Vertically-Resolved Cloud Amount for Joint CloudSat–CALIPSO Radar–Lidar Observations, Remote Sensing, 13, 807, https://doi.org/10.3390/rs13040807, 2021.

Marchand, R., Mace, G. G., Ackerman, T., and Stephens, G.: Hydrometeor Detection Using Cloudsat—An Earth-Orbiting 94-GHz Cloud Radar, Journal of Atmospheric and Oceanic Technology, 25, 519–533, https://doi.org/10.1175/2007JTECHA1006.1, 2008.

Marchand, R. and Mace, G.: Level 2 GEOPROF Product Process Description and Interface Control Document, Coop. Inst. for Res. in the Atmos., Fort Collins, Colo, https://www.cloudsat.cira.colostate.edu/data-products/2b-geoprof, 2018.

Mace, G. G. and Zhang, Q.: The CloudSat Radar-Lidar Geometrical Profile Product (RL-GeoProf): Updates, Improvements, and Selected Results: CLOUDSAT RADAR-LIDAR GEOMETRICAL PROFILE, Journal of Geophysical Research: Atmospheres, 119, 9441–9462, https://doi.org/10.1002/2013JD021374, 2014.

Winker, D. M., Hunt, W. H., and McGill, M. J.: Initial Performance Assessment of CALIOP, Geophysical Research Letters, 34, L19 803, https://doi.org/10.1029/2007GL030135, 2007.

**For clarity and utility, it might be beneficial to delineate exact numbers related to the uncertainties inherent in datasets across different global regions. Such specificity would aid users in gauging the reliability of the data in diverse contexts.**

We appreciate what reviewer #1 is asking for but we struggled to respond to this request. While it sounds attractive, we do not see a way to "delineate exact numbers" for all users and in the diverse contexts we see this product being used. These exact numbers will depend on the application of the dataset.

To help users in many diverse settings, our data product provides many easily-used and built in variables to quantify uncertainty, especially sampling uncertainty. See lines 182-188 and Table 1 for a description of these variables and Section 4.3 of the paper for a discussion of the sampling uncertainty and recommendations for users.

**One notable observation is the narrow focus on dataset evaluation at Utqiagvik, Alaska, despite its global ambit. Given the universal scope of the dataset, is there a reason for concentrating evaluation efforts in this isolated region? With resources like the ARM program, which boasts a spectrum of global observatories, a broader evaluation could reinforce the dataset's credibility across multiple geographies.**

We thank reviewer #1 for encouraging us to add more comparisons with ground-based sites for validation. We focused first on the Alaska site because of its frequent sampling by CloudSat+CALIPSO and ground-based remote sensing record covering the entire

CloudSat+CALIPSO mission. But we agree with reviewer #1 that additional sites could be added. In response to the reviewer, we explored other sites with more limited sampling. We found that the comparisons were of value for characterizing our data product. Thus, we have added three additional sites (Lamont, Oklahoma, USA; Graciosa Island, Azores; and Darwin, Australia) to the paper (Figure 10) and revised our discussion accordingly (lines 344-375).

*"We compare 3S-GEOPROF-COMB to four Atmospheric Radiation Measurement (ARM) ground sites, which offer cloud mask retrievals from a combination of ground-based radar and lidar (Xie et al., 2010). We choose ARM sites in Graciosa Island, Azores, Portugal in the Eastern North Atlantic (ENA C1); Utqiagvik, Alaska on the North Slope of Alaska (NSA C1, Verlinde et al. (2016)); Lamont, Oklahoma in the Southern Great Plains (SGP C1, Sisterson et al. (2016)); and Darwin, Australia in the Tropical Western Pacific (TWP C3, Long et al. (2016)). These sites represent a wide variety of cloud regimes for evaluation of our satellite-derived cloud product. Many studies have already compared surface-based radar/lidar and CloudSat/CALIPSO cloud measurements (e.g. Liu et al. (2010); Blanchard et al. (2014); Liu et al. (2017); Protat et al. (2014); Kim et al. (2008)), to which we refer the reader for more in-depth discussion. Our intent is to broadly compare multi-year climatologies from 3S-GEOPROF-COMB to explore the utility of our product.*

*We compare surface-derived cloud cover and vertical cloud fraction to 2.5° × 2.5° and 5° × 5° 3S-GEOPROF-COMB grid cells containing each ARM site (Fig. 10). Averages are calculated from all months where both surface and satellite observations are available (see Figure 5). The SGP and NSA comparisons are calculated from ~10 years of data, while the TWP and ENA comparisons are calculated from ~3.5 years of data (2006-2010 and 2015-2020, respectively). 3S-GEOPROF-COMB broadly captures the shape and magnitude of each location's seasonal cycle (Fig. 10, first row), where the greatest differences come from decreased satellite cloud cover in regions with more low clouds (ENA, Fig. 10a and NSA, Fig. 10d). If we exclude clouds with top heights below 500 m from the surface-based measurements, the region where surface clutter prevents CloudSat measurements, the agreement greatly improves (Fig. 10, first row, dash-dotted line). However, by excluding the region where surface clutter reduces CloudSat sensitivity (500-1000 m), the surface-derived cloud cover drops below satellite-derived at NSA (Fig. 10d) and to some extent ENA (Fig. 10a) (for other sites >500 m cover equals >1 km cover), suggesting that 3S-GEOPROF-COMB still contains relevant cloud information at altitudes 500-1000 m.*

*We also compare vertical cloud fraction for the month of best (Fig. 10, middle row) and worst (Fig. 10, last row) cloud over agreement at each site. 3S-GEOPROF-COMB generally shows more high clouds than the surface-based measurements, likely due to the difficulty of capturing optically thin clouds far from the surface (Kim et al., 2008; Protat et al., 2014). The product broadly captures the vertical cloud fraction for each month, though several comparisons have large (≥0.1) differences. The largest occurs in April at the TWP site (Fig. 10l), where satellite-derived cloud fraction is uniformly much larger than surface-derived cloud fraction. This difference likely comes from radar attenuation from the frequent heavy precipitation at the site (Long et al., 2016; Liu et al., 2010), where Liu et al. (2010) noted good ARM-CloudSat cloud*

*fraction agreement at the site for non-precipitating cases. Other points of note are January at ENA (Fig. 10a) and October at NSA (Fig. 10e), where the product does a good job of estimating cloud cover but overestimates the <1 km low cloud fraction peak. This is likely because of the finer vertical resolution (40 vs 240 m) of the surface-based instruments compared to 3S-GEOPROF-COMB, where the surface-based instruments spread the overall cloud occurrence across a wider array of height bins, thereby decreasing maximum cloud fraction."*

[Figure]

**Response to Reviewer #2**

**This manuscript describes a new data product which reports monthly, globally gridded 3-D cloud occurrence based on co-located observations from the CALIPSO and CloudSat satellites. The manuscript presents a comprehensive description of the product and a detailed comparison of the product with results from CALIPSO-only and CloudSat-only which helps in interpretation of the joint results. The manuscript does a good job of describing the product and its strengths and limitations, but the manuscript would benefit from clarification of a few points.**

We thank reviewer #2 for their positive review and constructive comments that have improved the manuscript.

**The discussion in lines 31-49 of detection sensitivities of radar and lidar is a little misleading in casting the comparison in terms of layer optical depth. The differences in detection sensitivity between the two instruments are primarily driven by particle size and not as much by concentration. Liquid clouds such as stratocumulus and altocumulus (composed of relatively small droplets) are often undetected by CloudSat even though they can be opaque to CALIOP. Small, cold ice particles detected by CALIOP are often missed by CloudSat, or CloudSat may only detect the drizzle component of these clouds. But CloudSat can detect low concentrations of very large ice particles (perhaps precipitating) which are not detectable by CALIOP due to the low concentrations. This is nicely characterized in Liu et al. (2016) and Liu et al. (2018). Another thing which is a little misleading: CloudSat surface clutter is due more to sidelobes of the radar beam than to the pulse length. CALIOP does not suffer from the same phenomenon not only because the laser has a much shorter pulse but it also has no sidelobes, due to the wavelength being three orders of magnitude shorter.**

We greatly appreciate the reviewer's guidance and expertise. We have revised our comparison of CPR and CALIOP to clarify the points raised. It now reads (lines 35-42):

*"Due to the lidar's shorter wavelength, scattering layers with small particle size and/or low optical thickness (e.g. aerosol or cirrus cloud layers) will have a stronger return for the lidar than the radar. While this increased sensitivity allows the lidar to detect thin cloud and aerosol layers, it also means that optically thick layers attenuate the lidar and prevent measurement below the level of attenuation (Liu et al., 2022). In contrast, while the radar does not detect optically thin layers or small droplet sizes, it only attenuates in the most extreme of precipitation events ($\sim$ 0.3% of profiles (Mace et al., 2007)). Second, the CloudSat radar has 'surface clutter' preventing measurement in the lowest 500 m of the atmosphere (Marchand et al. 2008), whereas CALIPSO's lidar allows measurement of clouds near the surface (Winker et al., 2009)."*

Additionally, we have mentioned the Liu et al. studies cited by the reviewer in our discussion of errors in our attenuation estimation (lines 332-334):

*"In particular, some attenuation differences may arise from warm marine clouds which are opaque to CALIPSO but go undetected by CloudSat (Liu et al., 2016, 2018), which Liu et al. (2018) found to be globally most prevalent over the Southern Ocean."*

**The sentence in lines 104-106 is a little confusing where it says that 2B-GEOPROF-LIDAR uses a "fixed maximum along-track averaging length of 5 km for cloud detection". This sounds like maybe 2B-GEOPROF-LIDAR is performing its own cloud detections using 5- km averaging. My understanding is that 2B-GEOPROF-LIDAR is using only the layers from the CALIPSO Layer Product which have been detected in 5-km averages and ignoring the layers in that product which are detected after additional averaging, but I could be wrong. This point should be clarified.**

We agree. We meant to present the 2B-GEOPROF-LIDAR averaging as the reviewer describes. We have revised the text (lines 104-105) to improve communication of this point:

*"The input cloud mask to 2B-GEOPROF-LIDAR uses along-track averaging of up to 80 km for cloud detection (Winker et al. 2009), but 2B-GEOPROF-LIDAR only considers clouds detected using 5 km of along-track averaging."*

**Line 130: The authors clearly explain here that 'lidar attenuation' is determined by comparing the CALIOP and CloudSat cloud masks. However, this may result in errors when CloudSat fails to detect a cloud which is opaque to CALIOP. This is likely responsible for some of the differences shown later in the paper. In a future version of this product, the authors might consider using the presence or absence of lidar surface detection to identify lidar full attenuation, which is the approach used for CALIOP Level 2 products.**

Unfortunately, the presence or absence of CALIOP surface return is not reported in 2B-GEOPROF-LIDAR.  As such, we have done our best to identify attenuation in 2B-GEOPROF-LIDAR using the variables that are available.  We requested CALIOP L2 attenuation information be added to 2B-GEOPROF-LIDAR for the next release (R06). If 2B-GEOPROF-LIDAR is updated to include CALIOP L2 attenuation, we will update our product to use it.

**Line 175: In partitioning cloud cover into High/Middle/Low levels the authors use a fixed altitude to approximate the ISCCP pressure levels, which introduces a source of uncertainty in comparisons between this product and results from passive sensors. It is straightforward to compute monthly zonal-mean altitudes corresponding to the ISCCP pressure levels, which mostly eliminates these uncertainties. This is the approach used for CALIPSO-ST and MISR in the GEWEX Cloud Assessment and authors should consider this approach for a future version of the data product.**

We are grateful for this useful suggestion and a straightforward means of implementing it. We have recalculated the data product with cloud level thresholds defined with monthly- and zonalmean 440 and 680 mb geometric heights from the NCEP-NCAR reanalysis. We have added a description of the process to the methodology section (lines 174-179):

*"We choose 680 mb and 440 mb as thresholds separating low, middle, and high cloud layers based on the International Satellite Cloud Climatology Project (ISCCP) (Rossow and Schiffer, 1999). Since our product is reported on height levels rather than pressure levels, we use the NCEP-NCAR reanalysis (Kalnay et al. 1996) to determine monthly- and zonal-mean 440 and 680 mb geometric heights for use as thresholds. The product applies 440 (680) mb height thresholds ranging from a minimum of 5.5 (2.5) at the poles to a maximum of 7 (3.5) km at the equator."*

**Line 200: "latitudes of the first and last DO-Op profiles" is mentioned here and in several places later in the manuscript. It wasn't clear to me what is meant by 'first' and 'last', please explain.**

We agree that more clarification is necessary.  We have added a more detailed presentation and definition of these terms when they are first used (lines 201-207):

*"Each orbit begins with a descending (southward) equator crossing in Earth's shadow. The satellite then enters sunlight over the Southern Hemisphere, after which the radar powers on and begins data collection. We call the latitude and branch (ascending/descending) at which this occurs the first DO-Op profile. The satellite then exits sunlight over the Northern Hemisphere and halts data collection, which we call the last DO-Op profile. Example orbits are shown in Figure 4c, where the portion of the orbit with (without) DO-Op data collection is shown as a solid (dashed) line. The portion of the orbit with DO-Op data collection varies systematically as a function of the day of year, which we leverage to implement our subsampling scheme."*

**Line 306: The statement that CAL-COS 'excludes marine low clouds detected only on 5 km averaging lengths' is not very clear. What is done for marine low clouds in CAL-COS is that only single-shot detections of shallow liquid clouds are used and all the results based on horizontal averaging are ignored. We have realized single-shot detection is sensitive enough to detect virtually all of the shallow liquid clouds and that using layers detected in 5-km averages tends to overestimate shallow cloud cover.**

We agree the sentence is not clear and needs to be re-worded. The sentence now reads (lines 325-327):

*"This increase is due the fact that CAL-COS excludes shallow marine liquid clouds detected using along-track averaging, which tends to overestimate cloud cover (NASA/LARC/SD/ASDC, 2019, Detailed Data Quality Summary)."*

**Minor comments**

**Line 38: For me "level of attenuation" is ambiguous, as 'level' could refer to signal magnitude. I think 'altitude of attenuation' is what is meant.**
Corrected

**Line 57: I did not find a Figure 4.2c. Should this be Figure 2c?**
Corrected